# Entropic Balance Conditions and Optimization of Distillation Column System

**DOI:** 10.3390/e23111468

**Published:** 2021-11-06

**Authors:** Alexander Balunov, Ivan Sukin, Anatoly Tsirlin

**Affiliations:** 1Cybernetics Department, Yaroslavl State Technical University, 150023 Yaroslavl, Russia; 2Department of Data Analysis and Machine Learning, Financial University under the Government of the Russian Federation, 125993 Moscow, Russia; iasukin@fa.ru; 3Ailamazyan Program Systems Institute of RAS, 152120 Pereslavl-Zalessky, Russia

**Keywords:** finite-time thermodynamics, distillation, irreversibility, admissible mode set, structure optimization, load distribution, separation order

## Abstract

The paper considers the limitation problem of the distillation column systems separating multicomponent mixtures with serial and parallel structures. The solution takes into account the irreversibility of processes. Using entropic balance conditions, the dependence of load on heat consumption is obtained for a binary distillation column. This dependence is parameterized through two characteristic coefficients–reversible efficiency and irreversibility factor. This dependence was used to solve problems of distribution of heat and raw material fluxes in parallel column structure and selection of optimal separation order in serial structure. The obtained results make it possible to estimate the minimum heat consumption for the separation of a given flow of raw materials, the maximum productivity, and efficiency of the system.

## 1. Introduction

There is widespread research devoted to the problems of optimal organization of distillation (see [1,2,3,4,5,6,7,8,9,10,11,12,13,14,15,16,17,18,19,20], etc.). Often (see [1], p. 540), “the calculation methods are based on phase equilibrium diagrams and reversible process parameters while introducing empirical corrections” and rules such as (see [2], p. 289): “The most volatile component must be separated first” or “The component with the highest concentration must be separated first”.

Meanwhile, there is a dependence between the efficiency of a binary distillation column in the reversible and irreversible process, and the column capacity is limited by irreversibility of the processes flowing into it. This limitation stems from the kinetics of heat exchange in the still and the dephlegmator and kinetics of mass transfer along the height between the counter flows of reflux and steam. The kinetic factors depend on the size of contact surfaces and the organization of processes on the plates, i.e., the design and dimensions of the column (see [21]).

The energy consumption for separation depends on the boiling temperatures of the components and the compositions of the column inlet and outlet flows. These compositions determine the reversible isothermal work of separation (Gibbs separation), which does not depend on the separation order. When only reversible factors are taken into account, the separation order is determined by the boiling temperatures of the components. Adding irreversibility into consideration through entropy balance equations makes it possible to estimate the effect of separation order on both heat consumption during heat exchange and heat loss during mass transfer, and to select the separation sequence corresponding to the minimum total heat consumption in the column cascade.

In [9,22], irreversibility is taken into account through kinetics of heat and mass exchange. It can be used to trace the influence of kinetic factors on column peak performance (productivity, heat consumption) to evaluate inevitable irreversible energy expenses, and to organize the separation process so that these expenses are possibly reduced. However, in these works, the irreversibility of mass transfer is based on its algorithmic form. This does not present the dependence of the peak performance on the heat consumption and kinetic factors explicitly.

In [23], a relation between binary rectification column capacity and heat consumption is obtained, irreversibility considered. In this case, the approach of finite-time thermodynamics was used [24,25,26,27]. It shows that, for the Onsager kinetics of heat and mass transfer, this relationship can be parameterized as a convex upward parabola. The parabola-defining components are obtained as functions of the parameters of the column and the mixture to be separated.

Further, these results will be used to solve the problem of optimal organization of parallel and serial column structures. Specifically, a cascade that separates from a mixture of three components the one with the boiling point *T*_1_, which is intermediate between the temperatures *T*_0_ and *T*_2_ of the volatile and high boiling components.

The following questions are to be answered:How to distribute raw material and heat flows in the parallel columns system so that for a given flow rate of separated mixture and composition of product flows, the total heat consumption would be minimal?What is the maximum capacity of the parallel columns for given compositions of raw materials and product flows at their outlets?In which order should the cascade of serial columns separate the flow of raw materials into fractions of a given composition with minimum total heat consumption?Given the design parameters of the columns, which of them is reasonable to use in the first stage in a serial separation?What is the maximum target product throughput of a cascade of consecutive columns?

The dependence of entropy production on kinetics of processes in a distillation column plays the central role in solving the listed problems.

The Section 2 is devoted to obtaining the dependence of the productivity of a single column on the heat input, parametrization of this dependence, and obtaining the characteristic factors as a function of the parameters of the column, separated mixture, and separation products.

The Section 3 studies a parallel structure system for two and for *n* columns. 

The Section 4 deals with sequential structure systems for three and for *n* columns. 

## 2. Limitations of a Binary Distillation Column

### 2.1. Key Assumptions

Let us consider a common design of the column with heat input at the still and output at the dephlegmator (Figure 1).

Let us list the key assumptions used in [23] when considering a column with a given capacity:-Mass transfer is equimolar-In each section, pressures and temperatures of steam and liquid flows are close to each other (they change from section to section)-Effects of diffusion between adjacent sections are negligibly small-Heat from the outgoing flows is transferred to the incoming flows, and irreversibility of this heat exchange can be neglected-The flow of the separated mixture as a liquid is fed at the boiling point to the column section with the reflux composition close to the composition of the flow-Properties of the mixture are close to those of ideal solutions and the separation is close to clear.

These works studied an idealized model of the packed column. Two main sources of irreversibility are considered in this model: heat transfer into the still and out of the dephlegmator, and mass transfer between steam and reflux over the height of the column. A plate column with a sufficiently large number of plates is also close to this model. Note that the effective mass transfer factor, if it is calculated from data of the operating column, indirectly takes into account internal diffusion, mixing on plates, etc.

Molar fractions of low-boiling component in the raw materials flow $x_{F}$ and in the flows separated from the still and the dephlegmator $x_{B}$ and $x_{D}$, shall be assumed as given, as well as the mixture components in the still $T_{B}$ and in the dephlegmator $T_{D}$, which are determined by their boiling points. The selection fraction of the upper product ε depends on the compositions of the input and output flows. The material balance of the column for the low-boiling component brings:(1)$$\varepsilon =\frac{x_{F}-x_{B}}{x_{D}-x_{B}}$$

### 2.2. Relationship between Heat Consumption and Column Capacity

Assuming that the mixtures are close to ideal solutions and neglecting the heat of mixing, the equations of energy and entropic balances bring:$$q_{+}-q_{-}+g_{F}h_{F}-g_{F}\varepsilon h_{D}-g_{F}(1-\varepsilon )h_{B}=0$$
(2)$$g_{F}s_{F}-g_{F}\varepsilon s_{D}-g_{F}(1-\varepsilon )s_{B}+\frac{q_{+}}{T_{B}}-\frac{q_{-}}{T_{D}}+\sigma =0$$

Here, σ>0 is entropy production in column, $h_{F}$, $h_{D}$, $h_{B}$, $s_{F}$, $s_{D}$, $s_{B}$ are molar enthalpies and entropies of flows of product, distillate, and still products accordingly and $q_{+}$, $q_{-}$ are heat flows supplied to column still and taken from column dephlegmator correspondingly.

Assuming that the column is thermally insulated, and the heat loss to the environment is much less than the heat flux expended on the separation, it can be assumed that $q_{+}=q_{-}=q$.

The entropic balance Equation (2) taking into account (1) leads to:(3)$$q=g_{F}\frac{T_{B}}{T_{B}-T_{D}}\left[\left(s_{F}T_{D}-h_{F}\right)-\varepsilon \left(s_{D}T_{D}-h_{D}\right)-(1-\varepsilon )\left(s_{B}T_{D}-h_{B}\right)\right]+\sigma \frac{T_{B}T_{D}}{T_{B}-T_{D}}=q_{+}^{0}+\sigma \frac{T_{B}T_{D}}{T_{B}-T_{D}}$$

The first summand in the right side of this expression, which is denoted by $q^{0}$, represents the heat consumption in the reversible process when the heat and mass transfer factors (column size) are however large. It depends only on parameters of input and output flows and is proportional to column capacity $g_{F}$; the second one corresponds to dissipative energy consumption.

Let us designate $C_{F}$, $C_{D}$, $C_{B}$ as heat capacities of flows of raw material, distillate, and still residue. Before entering the column, the feed flow normally passes through a regenerative heat exchanger where it contacts the product flow exiting the still. In this case, the feed flow temperature at the heat exchanger inlet is close to $T_{D}$. In the heat exchanger (it is assumed to be “reversible” for simplicity), the heat balance equation is fulfilled:$$C_{B}(1-\varepsilon )(T_{B}-T_{D})=C_{F}(T_{F}-T_{D})$$
so that the flow of the still residue leaves it at temperature $T_{D}$.

Since the difference $\left(h-T_{D}s\right)$ for each of the flows is equal to the molar free energy, i.e., to the chemical potential µ of the mixture at $T=T_{D}$, the Equation (3) of heat flow with productivity is:(4)$$q=g_{F}\frac{T_{B}}{T_{B}-T_{D}}\left[\varepsilon \mu_{D}(T_{D},x_{D})+(1-\varepsilon )\mu_{B}(T_{D},x_{B})-\mu_{F}(T_{D},x_{F})\right]+\sigma \frac{T_{B}T_{D}}{T_{B}-T_{D}}$$

Each of the chemical potentials is:(5)$$\mu_{i}(T,P,x_{i})=\mu_{i}^{0}(T,P)+RT\ln x_{i},\;i=F,\;D,\;B$$

Since the chemical potentials in each section of the column correspond to the same temperature and pressure, their difference for the vapor phase is:$$\mu_{1}(T,y^{0})-\mu_{1}(T,y)=RT\ln\frac{y^{0}}{y},\;\mu_{2}(T,1-y)-\mu_{2}(T,1-y^{0})=RT\ln\frac{1-y}{1-y^{0}}$$
where *y* is the concentration of low-boiling component in the vapor phase in every section of the column and $y^{0}$ is the equilibrium concentration in the same section.

The right part of Equation (4) can be expressed through the compositions of flows:(6)$$q=g_{F}\frac{T_{B}}{T_{B}-T_{D}}\left[A_{F}-\varepsilon A_{D}-(1-\varepsilon )A_{B}\right]+\sigma \frac{T_{B}T_{D}}{T_{B}-T_{D}}=\frac{g_{F}A_{G}}{\eta_{C}}+\frac{\sigma T_{D}}{\eta_{C}}$$

Here, $A_{i}=RT_{D}\left[x_{i}\ln x_{i}+(1-x_{i})\ln(1-x_{i})\right]$, (i=F, D, B) are the reversible work of separating one mole of *i*-th flow into pure components, and the expression in (6) in square brackets is the reversible Gibbs separation of one mole of feed flow with concentration $x_{F}$ into flows with concentrations $x_{D}$ and $x_{B}$ at temperature $T_{D}$. It is designated as $A_{G}$. The value $\eta_{C}=(1-T_{D}/T_{B})$ is the thermal efficiency of the column, which is similar to the Carnot efficiency. Equating entropy production in (6) to zero brings us a reversible estimate $q^{0}=g_{F}A_{G}/\eta_{C}$ of the heat input in distillation.

A reversible distillation column can be thought of as an ideal heat machine operating between heat reservoirs with temperatures $T_{B}$ and $T_{D}$ and generating a separation power of $p^{0}=g_{F}A_{G}$.

Let us solve Equation (6) with respect to $g_{F}$. The result is:(7)$$g_{F}=q\frac{\eta_{C}}{A_{G}}-\sigma (q,g_{F})\frac{T_{D}}{A_{G}}$$

For the second summand in this equality in [23], an estimate is obtained below.

### 2.3. Irreversible Energy Losses

Let heat flows in the still and the dephlegmator be proportional to temperature difference:$$q=rV=\beta_{B}(T_{+}-T_{B})=\beta_{D}(T_{D}-T_{-})$$
where *V* is steam leaving the still, *r* is the molar heat of vaporization, $\beta_{B}$ and $\beta_{D}$ are heat transfer factors proportional to the heat exchange surfaces, and $T_{+}$ and $T_{-}$ are the coolant and refrigerant temperatures accordingly.

Entropy production due to heat exchange in the still and in the dephlegmator is:(8)$$\sigma_{q}=q\left(\frac{1}{T_{B}}-\frac{1}{T_{+}}+\frac{1}{T_{-}}-\frac{1}{T_{D}}\right)=q^{2}\left(\frac{1}{\beta_{B}T_{B}T_{+}}+\frac{1}{\beta_{D}T_{D}T_{-}}\right)$$

Here, the temperatures $T_{B}$ and $T_{D}$ are assumed to be known. At a given heat flow, temperatures $T_{+}$ and $T_{-}$ depend on the selected values of temperature differences in the still and the dephlegmator. When substituting in (8), they also determine $\sigma_{q}$.

To calculate the entropy production in the mass transfer process, we will use a model corresponding to a cask column with countercurrent flow of steam and liquid in a mode close to displacement. The steam flow V=q/r at equimolar mass transfer does not change and is related to the reflux *L* through the equation:

in the top section of the column
(9)$$L_{D}=\frac{q}{r}-g_{D}$$in bottom section
(10)$$L_{B}=\frac{q}{r}+g_{B}$$

Considering that for binary distillation the concentrations of the high-boiling component in the liquid and vapor flows are 1−x and 1−y, respectively, and the driving force of the process is determined by the difference of the current concentration y(x) and the equilibrium concentration $y^{o}(x)$, entropy production associated with mass transfer is expressed through flows and chemical potentials as:(11)$$\sigma_{g}=R\int_{x_{B}}^{x_{D}}\frac{1}{T(x)}\left\{g_{1}(y,y^{0})\left[\mu_{1}(T,y^{0})-\mu_{1}(T,y)\right]+g_{2}(1-y,1-y^{0})\left[\mu_{2}(T,1-y)-\mu_{2}(T,1-y^{0})\right]\right\}dx$$
where $g_{j}$ and $\mu_{j}$ (j=1, 2) are mass exchange fluxes and chemical potentials of components.

Considering the kind of chemical potentials (5) and equimolarity of mass transfer $\left(g_{1}(y,y^{0})=-g_{2}(1-y,1-y^{0})=g\right)$, the Equation (11) is rewritten as:(12)$$\sigma_{g}=R\int_{x_{B}}^{x_{D}}g(y,y^{0})\ln\frac{y^{0}(1-y)}{y(1-y^{0})}dx$$

The equations of material balance for low-boiling component for top and bottom column parts are:$$\begin{array}{c} \frac{q}{r}y(x)-g_{D}x_{D}-L_{D}x=0,\\ L_{B}x-\frac{q}{r}y(x)-g_{B}x_{B}=0 \end{array}$$

Taking into account (9) and (10), the working lines are obtained for the top and bottom of the column after substitution: $g_{D}=g_{F}\varepsilon$, $g_{B}=g_{F}(1-\varepsilon )$
(13)$$y^{D}(x,\frac{q}{r},g_{F})=\left(1-\frac{g_{F}\varepsilon r}{q}\right)x+\frac{x_{D}g_{F}\varepsilon r}{q}$$
(14)$$y^{B}(x,\frac{q}{r},g_{F})=\left(1+\frac{g_{F}(1-\varepsilon )r}{q}\right)x+\frac{x_{B}g_{F}(1-\varepsilon )r}{q}$$

The equations lead to $y^{D}(x_{D})=x_{D}$, $y^{B}(x_{B})=x_{B}$, $y^{D}(x_{F})=y^{B}(x_{F})=y_{F}$, and $y_{F}-x_{F}=\frac{g_{D}r}{q}(x_{D}-x_{F})$.

Substituting Equations (13) and (14) into Equation (12) defines $\sigma_{g}(q,g_{F})$ for a given mass transfer law. From that result, one calculates the sum of integrals on the intervals from $x_{B}$ to $x_{F}$ when $y(x)=y^{B}(x,\frac{q}{r},g_{F})$ and from $x_{F}$ to $x_{D}$ when $y(x)=y^{D}(x,\frac{q}{r},g_{F})$, which is only possible numerically.

For results in analytical form, let us find an estimate of $\sigma_{g}$ below by taking the law of mass transfer proportional to the driving force:$$g(y,y^{0})=k\frac{\mu_{1}(T,y^{0})-\mu_{1}(T,y)}{T}$$
where k is the effective mass transfer factor and T is the temperature in the selected cross section of the column.

After eliminating the difference of chemical potentials through the flow $g(y,y^{0})$, Equation (12) is:(15)$$\sigma_{g}(q,g_{F})=\frac{2}{k}\int_{x_{B}}^{x_{D}}g^{2}(y,y^{0})dx$$

Here, the multiplier 2 is related to the equimolar flow of the high-boiling component from vapor to liquid.

Let us introduce the mean flow:$$\bar{g}=\frac{1}{x_{D}-x_{B}}\int_{x_{B}}^{x_{D}}g(y,y^{0})dx$$
and using (15) we obtain the estimation below for $\sigma_{g}$ (see [9]):$$\sigma_{g}\geq \frac{2(x_{D}-x_{B}){\bar{g}}^{2}}{k}$$

This inequality turns into an equation when the mass transfer flow varies little over the column height.

Steam flow rate along the column height is constant and for the total amount of low-boiling component transferred from liquid to steam, one obtains the condition of material balance:$$\int_{x_{B}}^{x_{D}}g(y,y^{0})dx=V\left[y^{D}(x_{D})-y^{B}(x_{B})\right]=\frac{q}{r}(x_{D}-x_{B})$$

Wherefore $\bar{g}=q/r$, a
$$\sigma_{g}\geq \frac{2(x_{D}-x_{B})q^{2}}{kr^{2}}$$

The right part of this equation is used to estimate below the irreversibility of mass transfer.

Putting the total entropy production into Equation (7) gives an estimate for the capacity of the binary distillation column:(16)$$g_{F}\leq bq-aq^{2}$$

At the same time, characteristic factors *a* and *b* depend on kinetics of process, composition, and properties of separable substances as:(17)$$a=\left[\frac{1}{\beta_{B}T_{B}T_{+}}+\frac{1}{\beta_{D}T_{D}T_{-}}+\frac{2(x_{D}-x_{B})}{kr^{2}}\right]\frac{T_{D}}{A_{G}}$$
(18)$$b=\frac{T_{B}-T_{D}}{T_{B}A_{G}}=\frac{\eta_{C}}{A_{G}}$$
Here, $\eta_{C}$ is analogous to the Carnot efficiency, if a distillation column is considered a machine converting heat energy into separation work. At clear separation, the concentration of volatile component in the still product in Equation (17) is zero and in the dephlegmator it is equal to one.

Let us designate the parameter *b* as the reversible efficiency and the parameter *a* as the irreversibility factor. The first one depends only on the properties of the separated mixture and composition of the flows. The second one also depends on the kinetics of the process, and thus on the design features of the column.

The maximum viable heat consumption and the marginal capacity are determined by the characteristic factors. The performance is maximum at the heat consumption:$$q^{0}=\frac{b}{2a}$$
and reaches
$$g_{F}^{m}=\frac{b^{2}}{4a}$$

The section from zero to the heat flow $q^{0}$ forms the working part within the boundary of all possible column modes. A further increase in heat flow due to an increase in dissipation leads to a decrease in performance. In the working area, the heat consumption depends on the performance as:$$q=\frac{b}{2a}-\sqrt{\frac{b^{2}}{4a^{2}}-\frac{g_{F}}{a}}$$

Let us introduce the parameter $z=q/q^{0}$—the degree of column load. Then, efficiency of binary distillation column (see (16))
$$\eta_{C}=\frac{g_{F}}{q}=b(1-0.5z)$$
is proportional to the reversible efficiency. At full load z=1, column efficiency for the chosen form of kinetics does not depend on irreversible factors and is 0.5b (half of reversible efficiency). The very value of the peak performance and its corresponding heat consumption, of course, depends on these factors. This fact corresponds to the well-known fact [28] that the cycle efficiency of irreversible thermal machine corresponding to its maximum power (Novikov–Curzon–Albourne efficiency) does not depend on kinetics of the heat exchange of the working body with sources, while the peak power of the machine depends on it.

The Equation (17) includes effective mass transfer factor *k*, which, like heat transfer factors $\beta_{B}$, $\beta_{D}$, depends on column design and size.

### 2.4. Calculation of Irreversibility Factor and Reversible Efficiency Based on Measurement Results

The values of the characteristic factors can be determined by simple measurement because applying only two factors is sufficient to solve numerous problems. The factors obtained in this way can be refined based on the properties of the mixtures to be separated or the requirements for the product composition change, heat, and mass transfer devices “age”, etc.

For two-column modes with heat flow rates $q_{1}$ and $q_{2}$ and performances corresponding to these flow rates, $g_{F1}$ and $g_{F2}$ in operation mode range ($q_{1}>q_{2}\Rightarrow g_{F1}>g_{F2}$), characteristic factors are determined by relations resulting directly from (16):(19)$$a=\frac{q_{1}g_{F2}-q_{2}g_{F1}}{q_{1}q_{2}(q_{1}-q_{2})}b=\frac{g_{F1}}{q_{1}}+aq_{1}$$

If the characteristic factors are found from experimental data, Equations (17) and (18) can be used to calculate the kinetic factors.

### 2.5. Dependence of Reflux Ratio on Column Load and Characteristic Factors

The reflux ratio *R*, which is the ratio of reflux flowing back into the column to the product selection from the dephlegmator, is an important control parameter. Let us express it through the characteristic parameters. This is carried out by making the material balance equation on inlet and outlet flows of the dephlegmator:$$V=L+g_{F}\varepsilon$$

Here, *L* is the returning reflux. After dividing this equation by the product flow leaving the dephlegmator and replacing the steam flow rate with *q* and the heat of vaporization with *r*, one obtains for the working area:(20)$$R=\frac{q(g_{F})}{g_{F}x_{F}r}-1=\frac{b-\sqrt{b^{2}-4ag_{F}}}{2ag_{F}x_{F}r}-1$$

If the reflux ratio is measured on an operating column, Formula (20) can be used to calculate the characteristic factor.

## 3. The Limits of the Parallel Structure

The parallel structure of distillation columns is a structure consuming a limited flow of thermal energy. At the inlet of each *i*-th column, mixture $g_{Fi}$ flows in. Its composition, as well as compositions of outlet flows leaving the column, is known. Let us denote the total raw material flow as $g_{F}$. The columns differ in their design parameters, the temperatures of the flows heating the mixture in the still and cooling it in the dephlegmator. Due to this, for each *i*-th column, the characteristic factors $a_{i}$, $b_{i}$ found by Equations (17) and (18) or calculated from experimental data (see (19)), are different.

Let us express the limits of such structure through the characteristic factors of the columns.

### 3.1. Reversible Efficiency and Maximum Performance

Let the distribution of raw material flows $g_{Fi}$ between columns be given, and $\gamma_{i}$ be the fraction of this flow from $g_{F}$. Then, the heat consumption in the reversible mode and the reversible efficiency for the parallel structure are equal:$$q_{\Sigma }=\sum_{i}q_{i}=g_{F}\sum_{i}\frac{\gamma_{i}}{b_{i}},b=\frac{g_{F}}{q_{\Sigma }}=\frac{1}{\sum \left(\gamma_{i}/b_{i}\right)}$$

Maximum performance of the parallel structure and its corresponding heat consumption taking irreversibility into account:$$g_{F}^{\max}=\sum_{i}\frac{b_{i}^{2}}{4a_{i}},q_{\Sigma }^{\max}=\sum_{i}\frac{b_{i}}{2a_{i}}$$

### 3.2. Optimal Distribution of Heat Flows

Let us consider the problem of such distribution of heat flow between the columns, so that for a given total heat consumption $q_{\Sigma }$, the productivity *n* of parallel columns $g_{F}$ would be maximal. The problem takes the form:(21)$$g_{F}=\sum_{i}\left(b_{i}q_{i}-a_{i}q_{i}^{2}\right)\to \max,\;/\sum_{i}q_{i}=q_{\Sigma }<q_{\Sigma }^{\max}$$

Since at the working part of the load parameter $g_{Fi}(q_{i})$ is monotonous and convex upwards, Equation (21) has the same solution as the dual problem about the minimum heat consumption at a given performance.

The Lagrange function of Equation (21) for the nondegenerate solution is:$$L=\sum_{i}\left(b_{i}q_{i}-a_{i}q_{i}^{2}-\lambda q_{i}\right)$$

Its stationarity conditions on $q_{i}$:$$b_{i}-2a_{i}q_{i}=\lambda ,i=1,\;2,\;\ldots \;,n$$

After substituting $q_{i}=\frac{b_{i}-\lambda }{2a_{i}}$ into (21) and excluding λ from the condition of given total heat flow, one obtains optimal distribution of heat and raw material flows between columns:(22)$$q_{i}^{*}=\frac{b_{i}}{2a_{i}}-\frac{\sum_{j}\frac{b_{j}}{2a_{j}}-q_{\Sigma }}{a_{i}\sum_{j}\frac{1}{a_{j}}},\;g_{i}^{*}=b_{i}q_{i}^{*}-a_{i}q_{i}^{*2},\;i,\;j=1,\;2,\;\ldots \;,n$$

## 4. Limits of the Serial Structure

A cascade of several columns connected in series or several such parallel cascades is used to separate multi-component mixtures into individual components or into fractions consisting of several components. The resulting structure goes through the optimization of each of the parallel cascades and their optimal matching as was demonstrated in the previous section.

Initially, consider in detail a cascade of two consecutive columns in which the mixture must be separated into three components or three fractions. In particular, this task arises when a single component or fraction with intermediate boiling point properties is isolated from a mixture.

### 4.1. The Possible Modes and Choice of Sequence of Separation of a Three-Component Mixture

After representing column load dependence on heat consumption through characteristic factors, they can be used to optimize a cascade of two columns by finding their values by Equations (17) and (18) or by experimental data.

Let us choose the separation sequence by minimum total heat consumption at a given performance and composition of the flows. As above, a clear separation in each column is assumed.

Let the components of the mixture be ordered and marked with the indices 0, 1, and 2. Their molar concentrations in the flow of the separated mixture $x_{0}$, $x_{1}$, $x_{2}$ and boiling points $T_{0}<T_{1}<T_{2}$ are given. Let us introduce notations for characteristic coefficients of each of the columns at each of the separation orders.

For the direct order, when in the first column the zeroth component is separated, and in the second column the first and second components are separated, let us denote the characteristic factors in the parametric representation of each column as the index *d*. For example, $b_{d1}$ is the reversible efficiency at direct separation order for the first column.

For the reverse order, where the second component is separated in the first column, and the remaining mixture is divided in the second column, $b_{r1}$ is the reversible efficiency for the first column and $b_{r2}$ for the second column.

Let the molar heat of vaporization of low-boiling $r_{0}$ and medium-boiling $r_{1}$ components be known. The molar heat of vaporization of a mixture of low-boiling and medium-boiling components is a weighted average $r_{01}=\left(r_{0}x_{0}+r_{1}x_{1}\right)/\left(x_{0}+x_{1}\right)$.

In the direct separation order on the first column $T_{D}=T_{0}$, and the temperature in the still, where the first and second components are, is close to the boiling point of the first $T_{B}=T_{1}$. In the second column: $T_{D}=T_{1}$, $T_{B}=T_{2}$.

In reverse order of separation for the first column: $T_{D}=T_{1}$, $T_{B}=T_{2}$. For the second column: $T_{D}=T_{0}$, $T_{B}=T_{1}$.

The flowsheet for the direct separation order is shown in Figure 2. The flowsheet for the reverse order of separation has a similar structure. 

Initially, it leads to an inequality defining the condition at which the total reversible heat consumption for the direct separation order is less than for the reverse one. Then, the problem on the boundary of achievable modes of the cascade is solved taking irreversibility into account. This solution determines the optimal order of separation taking irreversible factors into account.

### 4.2. Selection of the Separation Order in the Reversible Approximation

Reversible efficiencies of columns for direct order of separation according to (18) are:(23)$$b_{d1}=-\frac{T_{1}-T_{0}}{RT_{1}T_{0}\left(x_{0}\ln x_{0}+(1-x_{0})\ln(1-x_{0})\right)},\;b_{d2}=-\frac{T_{2}-T_{1}}{RT_{2}T_{1}\left(x_{1}\ln x_{1}+x_{2}\ln x_{2}-(x_{1}+x_{2})\ln(x_{1}+x_{2})\right)}$$

Similarly, for the reverse order of separation:(24)$$b_{r1}=-\frac{T_{2}-T_{1}}{RT_{2}T_{1}\left(x_{2}\ln x_{2}+(1-x_{2})\ln(1-x_{2})\right)},\;b_{r2}=-\frac{T_{1}-T_{0}}{RT_{1}T_{0}\left(x_{0}\ln x_{0}+x_{1}\ln x_{1}-(x_{0}+x_{1})\ln(x_{0}+x_{1})\right)}$$

In these two expressions, the efficiency is the ratio of flow at the inlet to the cascade per unit of heat consumed, taking into account that the second column receives a flow equal to $g_{F}(x_{1}+x_{2})$ for the direct and $g_{F}(x_{0}+x_{1})$ for the reverse separation order.

The magnitude of the separable flow has no effect on the reversible parameters, nor does the universal gas constant *R* in (23) and (24). The inequality defining the condition under which the direct order of separation corresponds to a lower heat input is:(25)$$\left(1/b_{d1}+1/b_{d2}\right)<\left(1/b_{r1}+1/b_{r2}\right)$$

After some simple calculations, it leads to:$$K_{1}\left(x_{1}\ln x_{1}-(1-x_{2})\ln(1-x_{2})-(1-x_{0})\ln(1-x_{0})\right)<\;<K_{2}\left(x_{1}\ln x_{1}-(1-x_{0})\ln(1-x_{0})-(1-x_{2})\ln(1-x_{2})\right)$$
or
$$K_{1}=\frac{T_{1}T_{0}}{T_{1}-T_{0}}<K_{2}=\frac{T_{2}T_{1}}{T_{2}-T_{1}}$$

### 4.3. The Realizable Modes of Cascade Separation of Three-Component Mixture

Let us write down the relations defining the realized modes of cascade through characteristic parameters of columns at direct and reverse order of separation, taking irreversible factors into account. The peak heat flow and the peak performance of the column are inversely proportional to *a*. Since the raw material flow to the first column is greater than to the second column, it is advisable to put the column with less irreversibility factor at the selected separation order first.

The peak performance of the cascade $g_{{}_{F}}^{*}$ depends on the mixture separation order:

for direct order
(26)$$g_{Fd}^{*}=\min\left(\frac{b_{d1}^{2}}{4a_{d1}};\;\frac{b_{d2}^{2}}{4a_{d2}(1-x_{0})}\right)$$for reverse order
(27)$$g_{Fr}^{*}=\min\left(\frac{b_{r1}^{2}}{4a_{r1}};\;\frac{b_{r2}^{2}}{4a_{r2}(1-x_{2})}\right)$$

For the direct separation order, the relations defining $q_{d1}$, $q_{d2}$, and $q_{d}$ for a given mixture flow rate:(28)$$q_{d1}=\frac{b_{d1}}{2a_{d1}}-\sqrt{\frac{b_{d1}^{2}}{4a_{d1}}-\frac{g_{F}}{a_{d1}}},q_{d2}=\frac{b_{d2}}{2a_{d2}}-\sqrt{\frac{b_{d2}^{2}}{4a_{d2}}-\frac{g_{F}(1-x_{0})}{a_{d2}}},q_{d}=q_{d1}+q_{d2}$$

The same for the reverse order of separation:(29)$$q_{r1}=\frac{b_{r1}}{2a_{r1}}-\sqrt{\frac{b_{r1}^{2}}{4a_{r1}}-\frac{g_{F}}{a_{r1}}},q_{r2}=\frac{b_{r2}}{2a_{r2}}-\sqrt{\frac{b_{r2}^{2}}{4a_{r2}}-\frac{g_{F}(1-x_{2})}{a_{r2}}},q_{r}=q_{r1}+q_{r2}$$

These consumptions are constrained by inequalities:$$q_{d1}\leq \frac{b_{d1}}{2a_{d1}},q_{d2}\leq \frac{b_{d2}}{2a_{d2}},q_{r1}\leq \frac{b_{r1}}{2a_{r1}},q_{r2}\leq \frac{b_{r2}}{2a_{r2}}$$

The optimal order of separation for which the heat consumption is less for a given performance leads to the inequality
(30)$$q_{d1}+q_{d2}<q_{r1}+q_{r2}$$
determining the choice of the direct separation order taking irreversibility into account. If the right side of this inequality is larger than the left side, then the optimal order is reversed.

At the working section, the cascade performance monotonically depends on total heat consumption. Therefore, the solution to the problem of minimum heat consumption at given performance coincides with the solution to the problem of maximum performance at given heat consumption.

Note that the left and right sides of the Equation (30) through the Equations (28) and (29) depend on the flow rate $g_{F}$ of the separated mixture, and it may turn out that at one capacity there is more sense in using the direct, and at the other—the reverse order of separation.

The peak performance for each of the products is $x_{i}g_{F}^{*}$, i=0, 1, 2.

### 4.4. Example of Selecting a Separation Order

Here is an example of the column cascade calculation, using the above-obtained relations.

1.Source data.

Component concentrations and their boiling points:$$x_{0}=0.5,\;x_{1}=0.3,\;x_{2}=0.2,\;T_{0}=393\;K,\;T_{1}=438\;K,\;T_{2}=458\;K$$

Vaporization heat of two volatile components and their mixture:$$r_{0}=50,000\;J/\mathrm{mol},\;r_{1}=70,000\;J/\mathrm{mol},\;r_{01}=57,500\;J/\mathrm{mol}$$

Mass and heat transfer rates in the columns for both separation options:$$k_{d1}=13\;({\mathrm{mol}}^{2}\cdot K)/(J\cdot s),\;k_{d2}=11\;({\mathrm{mol}}^{2}\cdot K)/(J\cdot s),\;k_{r1}=15\;({\mathrm{mol}}^{2}\cdot K)/(J\cdot s),\;k_{r2}=13\;({\mathrm{mol}}^{2}\cdot K)/(J\cdot s),\;\;\beta_{1}^{B}=70,000\;W/K,\;\beta_{2}^{B}=20,000\;W/K,\;\beta_{1}^{D}=75,000\;W/K,\;\beta_{2}^{D}=22,000\;W/K$$

Required performance $g_{F}$ = 1 mol/s.

2.Let us calculate the characteristic parameters for each column.

Reversible mixture separation work:$$A_{Gd1}=-RT_{0}\left[x_{0}\ln x_{0}+(1-x_{0})\ln(1-x_{0})\right]=2258\;J/\mathrm{mol},\;A_{Gd2}=1223\;J/\mathrm{mol},\;A_{Gr1}=1819\;J/\mathrm{mol},\;A_{Gr2}=1720\;J/\mathrm{mol}$$

Reversible efficiencies by Equations (20) and (21):$$b_{d1}\;=4.55\cdot 10^{-5}\;\mathrm{mol}/J,\;b_{d2}\;=3.57\cdot 10^{-4}\;\mathrm{mol}/J,\;\;b_{r1}=2.40\cdot 10^{-5}\;\mathrm{mol}/J,\;b_{r2}=5.97\cdot 10^{-5}\;\mathrm{mol}/J$$

The irreversibility factors for the direct and reverse separation order are calculated by Equation (17), considering the temperatures $T_{+}$ and $T_{-}$ to be close to $T_{B}$ and $T_{D}$, respectively:$$a_{d1}=3.38\cdot 10^{-11}\;(\mathrm{mol}\cdot s/J^{2}),\;a_{d2}=18.0\cdot 10^{-11}\;(\mathrm{mol}\cdot s/J^{2}),\;a_{r1}=4.27\cdot 10^{-11}\;(\mathrm{mol}\cdot s/J^{2}),\;a_{r2}=13.3\cdot 10^{-11}\;(\mathrm{mol}\cdot s/J^{2})$$

3.Equation (25), which defines the separation order in the reversible approximation, is accurate up to a constant multiplier 10^5^:(1/4.55+1/3.57)<(1/2.4+1/5.97)
It is fulfilled; hence the direct order is preferable.

4.Maximum cascade performance for direct and reverse separation order by Equations (26) and (27):$$g_{F}^{*d}=\min\left[\frac{{\left(4.55\right)}^{2}10}{43.38};\;\frac{{\left(3.57\right)}^{2}10}{4180.5}\right]=3.54\;\mathrm{mol}/s,\;g_{F}^{*r}=\min\left[\frac{{\left(2.4\right)}^{2}10}{44.27};\;\frac{{\left(5.97\right)}^{2}10}{4130.8}\right]=3.37\;\mathrm{mol}/s$$

The given performance is less than the maximum capacity for both separation options.

5.For each option, the total heat consumption by Equations (28) and (29) for $g_{F}$ = 1 mol/s are:$$q_{d1}=23\;\mathrm{KW},\;q_{d2}=16\;\mathrm{KW},\;q_{d}=39\;\mathrm{KW},\;q_{r1}=45.2\;\mathrm{KW},\;q_{r2}=157.5\;\mathrm{KW},\;q_{r}=202.7\;\mathrm{KW}.$$

Therefore, according to (30) and taking irreversibility into account, the direct order of separation is preferable.

### 4.5. Generalization for a Multi-Component Mixture

The results obtained can be generalized to the case of mixture separation of *n* components with concentrations each equal to $x_{i}$ into three fractions with concentrations $X_{0}=\sum_{0}^{\nu }x_{i}$, $X_{1}=\sum_{v+1}^{j}x_{i}$, $X_{2}=\sum_{j+1}^{n}x_{i}$ assuming clear separation.

For the mixture in the dephlegmator to contain all components of the separable fraction, its temperature must be equal to the highest boiling point of the component of this fraction. Similarly, for the liquid in the still to contain all the components of the fraction being separated into the still, its temperature cannot be higher than the minimum boiling point of the component fraction in the still. As a consequence, all results obtained above are valid after replacing the concentrations of components with concentrations of fractions and calculation of temperature coefficients by formulas:$$K_{1}=\frac{T_{\nu +1}T_{\nu }}{T_{\nu +1}-T_{\nu }},K_{2}=\frac{T_{j+1}T_{j}}{T_{j+1}-T_{j}}$$

Due to this, there will be no further distinctions for separation tasks into components and into fractions.

### 4.6. Case of n Consecutive Columns

More than two columns are connected in series when in each column one of the outlet flows does not require further separation and the second flow enters the next cascade column for separation. If both flows are separated in two subsequent columns, it is the parallel structure discussed above.

Let us focus on a consistent structure. When separating a mixture of *n* components, the cascade consists of (*n* − 1)-th column. The columns are selected so that their respective irreversibility factors increase as the flow rate of the mixture decreases along the separation process.

The Bellman dynamic programming algorithm [29] can be used to optimize this structure as follows:For the last two columns, all possible fractions of the three components with close boiling points (*n* − 2) are combed through. For each such fraction, using Equations (23)–(26), one finds the optimal separation order and the corresponding minimum heat consumption.One adds to this cascade the third to last column and for the resulting cascade chooses all fractions from the four components having similar boiling points (*n* − 3). Of the two possible options for direct and reverse order of separation of each such fraction, the best one is selected, taking into account that each order of separation corresponds to the optimal performance of the cascade of the last two columns. Thus, each mixture of four closely related components is assigned the order of its separation and the minimum required heat consumption.The fourth to last column of the cascade is added, and all five-component fractions are screened in the same way.

The calculation ends when the fraction size becomes *n* − 1, and there are only two variants. 

The performance of the column sequence is limited by the maximum performance of any one column. It is reasonable to choose the column with the maximum $\frac{b_{i}^{2}}{4a_{i}}$ at *i* as the first cascade column. This fraction defines the maximum raw material flow performance.

## 5. Conclusions

The results of the research show that to optimize structures consisting of several distillation columns, one can use the parametrized representation of performance dependence of each column on heat consumption. The entropic balance equation was used to derive this relationship. The characteristic factors can be calculated either from mixture characteristics and kinetic factors or directly from experimental data. The relation of reflux ratio with characteristic factors was obtained.

The research also dealt with the problem of load sharing for the parallel structure and the choice of separation order for the serial structure. The results make it possible to optimize complex systems by breaking them into subsystems, each of which can be represented as a parallel or sequential structure.

## Figures and Tables

**Figure 1 entropy-23-01468-f001:**
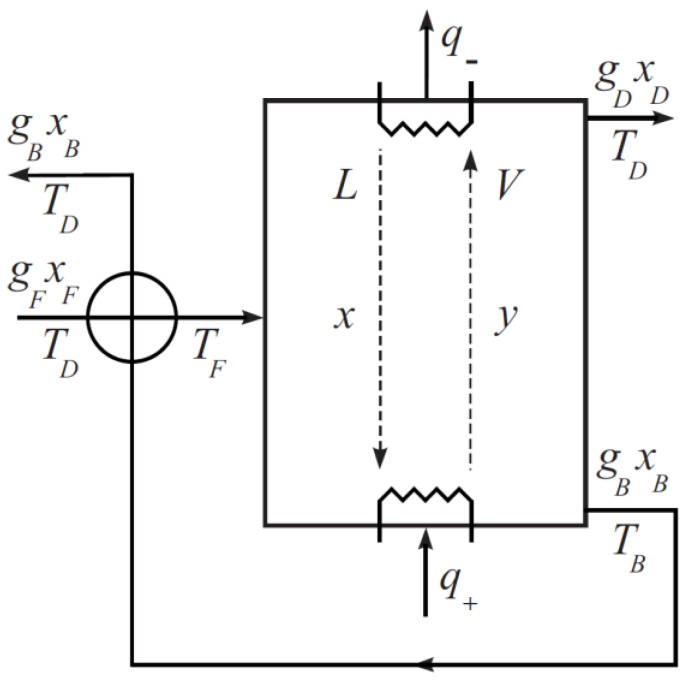
Calculation scheme of binary distillation column.

**Figure 2 entropy-23-01468-f002:**
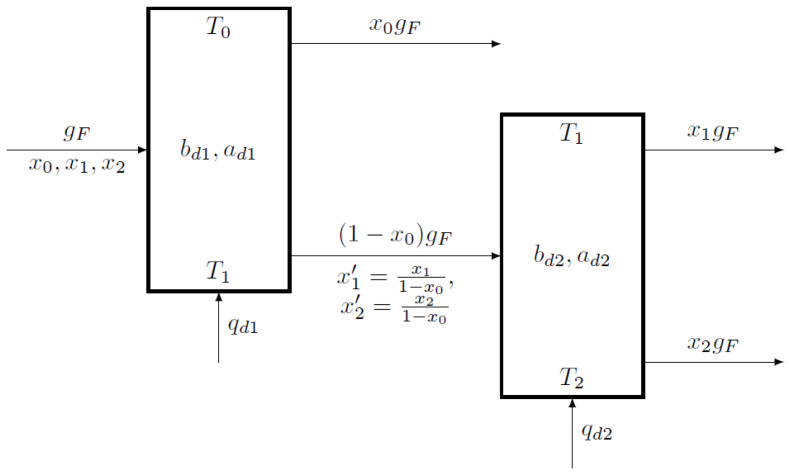
Distillation column cascade flowsheet for the direct separation order.

## Data Availability

All of the data is contained in the paper.

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
