# Peer review of "Entropic Balance Conditions and Optimization of Distillation Column System"

_entropy, 2021, doi:10.3390/e23111468_

Round 1

Reviewer 1 Report

  1. The article title contains “distillation column” but the article contains only a dephlegmator.
  2. The figure 1 presents a depglegmator and an exchanger. The output temperature of bottom product, TD, is equal with temperature of feed column. This phenomena is not possible.
  3. The mathematical models developed by authors are very complex but the presentation of these models is por. The presentation will be structured in:
  • The scope of model;
  • Input data;
  • Demonstration;
  • Final form of model;
  • Numerical exemple.
  1. Line no 170 is unclear.
  2. For paragraph 4.2 is necessary a principial scheme of the column structure.

6. For paragraph 4.4 is necessary to define the chemical components of the mixure and a principial scheme of the columns structure.

Author Response

Thank you for your valuable review!

  1. We cannot agree with this comment. The manuscript considers an irreversible distillation column and systems of such columns. Dephlegmator is only a part of such system and we do not consider it on its own.
  2. The Fig. 1 shows the structure of a column as a whole. The temperature of the bottoms is not . This flux as it is shown at Fig. 1 is being cooled in the regenerative heat exchanger down to the temperature , while heating the feed flux up to the temperature . The fact that the inlet temperature of this heat exchanger is  follows from the following simplifying considerations:
    If the heat capacities of the material fluxes in the heat exchanger are equal and the boiling point at the feed depends linearly on the ratio of the bottoms and distillate fluxes. The feed flux could always be set to unity. So, we have the following energy balance for the heat exchanger:

    where  is the ratio of the distillate flux to the feed flux and  is the temperature of the bottoms flux at the outlet of the heat exchanger. Solution these equations for  gives us .
    This equality is exact when assumptions mentioned above are satisfied.
  3. The mathematical model of a single distillation column has been a subject of the variety of papers. Some of these papers are mentioned in the References section. They also contain calculation methodology. We wanted to show how to use the parameterized model of a distillation column for optimization and estimation of the maximum capacity of the system of distillation columns.
  4. Thank you, we have corrected it.
  5. The structure of a single column is shown at the Fig. 1. We have added Fig. 2 showing the flowsheet of the two-column cascade.
  6. 2 shows the flowsheet of the two-column cascade. Key variables corresponding to the columns are shown at this figure.

Reviewer 2 Report

The excellent paper

Author Response

Thank you for the review!

The Notation section is quite large, so we think it's better to leave it at the end of the manuscript.

Reviewer 3 Report

see the attached

Author Response

Thank you for the review!

  1. We have added "finite-time thermodynamics" to the keywords
  2. and 3. We have extended the References section of the manuscript.

Round 2

Reviewer 1 Report

This new form of the article is better than old form.